# Complete Life Cycle of *Trypanosoma thomasbancrofti*, an Avian Trypanosome Transmitted by Culicine Mosquitoes

**DOI:** 10.3390/microorganisms9102101

**Published:** 2021-10-05

**Authors:** Magdaléna Fialová, Anežka Santolíková, Anna Brotánková, Jana Brzoňová, Milena Svobodová

**Affiliations:** Department of Parasitology, Faculty of Science, Charles University, 28 44 Prague, Czech Republic; asantolikova@gmail.com (A.S.); brotana@natur.cuni.cz (A.B.); jana.brzonova@natur.cuni.cz (J.B.)

**Keywords:** avian parasite, *Trypanosoma thomasbancrofti*, Culex, mosquito, life cycle, transmission, prediuresis

## Abstract

Avian trypanosomes are cosmopolitan and common protozoan parasites of birds; nevertheless, knowledge of their life cycles and vectors remains incomplete. Mosquitoes have been confirmed as vectors of *Trypanosoma culicavium* and suggested as vectors of *T. thomasbancrofti*; however, transmission has been experimentally confirmed only for the former species. This study aims to confirm the experimental transmission of *T. thomasbancrofti* to birds and its localization in vectors. *Culex pipiens* were fed on blood using four strains of *T. thomasbancrofti*, isolated from vectors and avian hosts; all strains established infections, and three of them were able to develop high infection rates in mosquitoes. The infection rate of the culicine isolates was 5–28% for CUL15 and 48–81% for CUL98, 67–92% for isolate OF19 from hippoboscid fly, while the avian isolate PAS343 ranged between 48% and 92%, and heavy infections were detected in 90% of positive females. Contrary to *T. culicavium*, trypanosomes were localized in the hindgut, where they formed rosettes with the occurrence of free epimastigotes in the hindgut and midgut during late infections. Parasites occurred in urine droplets produced during mosquito prediuresis. Transmission to birds was achieved by the ingestion of mosquito guts containing trypanosomes and via the conjunctiva. Bird infection was proven by blood cultivation and xenodiagnosis; mature infections were present in the dissected guts of 24–26% of mosquitoes fed on infected birds. The prevalence of *T. thomasbancrofti* in vectors in nature and in avian populations is discussed in this paper. This study confirms the vectorial capacity of culicine mosquitoes for *T. thomasbancrofti*, a trypanosome related to *T. avium*, and suggests that prediuresis might be an effective mode of trypanosome transmission.

## 1. Introduction

Digenetic protozoa of the genus *Trypanosoma* (Euglenozoa; Kinetoplastea; Trypanosomatida) [1] are blood parasites transmitted by bloodsucking invertebrates, notorious for the illnesses they cause in humans and animals (Chagas disease, sleeping sickness, nagana, etc.). Trypanosomes were found in birds more than 120 years ago by Danilewsky [2]. Recently, they were divided into three paraphyletic groups named after principal species: *Trypanosoma avium, Trypanosoma culicavium*/*Trypanosoma corvi*, and *Trypanosoma bennetti* [3]. Even though avian trypanosomes are widespread and their prevalence in birds can be high [4,5,6,7,8,9], they are neglected due to their low pathogenicity and economic importance. 

Knowledge of avian trypanosomes’ vectors remains incomplete, despite their importance in parasite life cycles. The diversity of trypanosome dipteran vectors is high; namely, *T. avium* s. s. is transmitted by blackflies (Simuliidae) [10,11,12] and sandflies (Psychodidae) [13,14], *T. corvi* by hippoboscid flies (Hippoboscidae) [15], *T. bennetti* group by biting midges (Ceratopogonidae) [16,17,18], and *T. culicavium* by mosquitoes (Culicidae) [19]. The mode of transmission into birds is by ingestion of the infected vector [12,15,19] or via the conjunctiva [12]. 

Mosquitoes were among the first studied vectors of avian trypanosomes in times when molecular barcoding was not available [10,20,21,22]. Until recently, *T. culicavium* was the only species of avian trypanosome transmitted by mosquitoes whose life cycle had been experimentally confirmed. Mature *T. culicavium* infections are localized strictly on the stomodeal valve [19], while other mosquito infections are found in the hindgut or midgut [10,21,23]. Recently, *T. thomasbancrofti* was described from the regent honeyeater (*Anthochaera phrygia*), a passerine endemic to Australia [24]. This species is identical to *T. avium* s. l. lineage II, which contained isolates originating from chiffchaffs (*Phylloscopus collybita*) and the mosquito *Culex pipiens* from Czechia [3]. Based on the high similarity of SSU rRNA sequences from Australian and Czech trypanosomes, *T. thomasbancrofti* was described as cosmopolitan, with the mosquito *Culex pipiens* being the suspected vector [24]. *Trypanosoma thomasbancrofti* was localized in the hindgut of the mosquito [3,22]. Since we possessed several isolates belonging to *T. thomasbancrofti*, we were interested in the life cycle, and transmission mechanism of a mosquito trypanosome species, which was probably among the first found in vectors [10,21]. Data on the natural prevalences in mosquitoes and birds are given as well.

## 2. Materials and Methods

### 2.1. Parasite Strains and Cultures

All *T. thomasbancrofti* strains used in this study were our own isolates originating from Czechia: CUL15 was isolated from a *Culex pipiens* female from Prague, Central Bohemia (ICUL/CZ/2000/CUL15); CUL98 from a *Culex pipiens* from Milovice forest, South Moravia (ICUL/CZ/2018/CUL98); OF19 from a hippoboscid fly *Ornithomya fringillina* from Neuměřice, Central Bohemia (IORN/CZ/2016/OF19); and PAS343 from a wood warbler (*Phylloscopus*
*sibilatrix*) from Milovice forest, South Moravia (APHY/CZ/2016/PAS343).

Trypanosomes were cultivated on rabbit (Bioveta, Ivanovice na Hané, Czech republic) or sheep (LabMediaServis, Jaroměř, Czech republic) blood agar (SNB-9), in flat tubes with liquid medium containing RPMI 1640 (Sigma-Aldrich, St. Louis, MO, USA) and Schneider’s Drosophila Medium (Sigma-Aldrich) mixed 1:1, supplemented with 10% FCS (Gibco, Thermo Fisher Scientific, Inc., Waltham, MA, USA), 2% sterile human urine, and 50 µg/mL amikacin (Medochemie, Prague, Czech republic).

### 2.2. Experimental Infections of Mosquitoes

Mosquitoes were bred in our laboratory: *Culex pipiens quinquefasciatus* (*Cx. quinquefasciatus* henceforth) originating from India, kept in our laboratory for more than 30 years, and *Culex pipiens molestus* (*Cx. molestus* henceforth) that were colonized recently (2016). Mosquito females were infected by feeding through a chick skin membrane on heat-inactivated (30 min in 56 °C) rabbit or sheep blood with 12–18 day-old culture of 2–6 × 10^8^ parasite cells/mL. Fed females were separated after blood feeding, kept at 21 °C, 60% humidity, and with access to 50% sucrose solution on a cotton pad. Mosquitoes were dissected 10–27 days post-infection, and their guts were examined under the light microscope for infection status, intensity, parasite localization, and its dynamics. Infection intensities were defined as: light, 1–100 parasites; medium, 100–1000 parasites; and heavy, >1000 parasites per gut. 

Experimental infections were always performed in pairs, both subspecies of *Culex* mosquitoes feeding at the same time on the same blood-parasite cocktail to minimize the influence of experimental conditions. Experiments were repeated three times with strain PAS343, twice with strains CUL15 and OF19, while strain CUL98 was tested only once.

### 2.3. Experimental Inoculation of Birds

Four canaries (*Serinus canaria*) and two zebra finches (*Taeniopygia guttata*) were screened before the experiment by blood cultivation as described in [14] from the metatarsus vein articulation (*vena metatarsalis plantaris superficialis media*) for trypanosome infections; all were negative. Birds were inoculated with 7–8 *Cx. molestus* or *Cx. quinquefasciatus* guts infected with different strains of *T. thomasbancrofti* (CUL98, OF19, PAS343), which were homogenized in saline, applied orally or placed on the conjunctiva. Infection status was checked by blood cultivation at 7–14 day intervals; cultures were checked 3 times in weekly intervals. In positive cases, parasite identity was confirmed by sequencing of the SSU rRNA. 

### 2.4. Transmission of Trypanosomes from Canaries to Mosquitoes

Mosquitoes were allowed to feed on trypanosome-positive birds to test if trypanosomes could establish infection in mosquitoes after natural exposure. Positive birds were kept for 60 min in a small cage placed into the net with 50 mosquito females in complete darkness. Blood-fed mosquitoes were separated, kept at 21 °C, 60% humidity, with access to 50% sucrose solution on a cotton pad, and dissected 10–15 days after feeding. Guts were examined under the light microscope for infection status.

### 2.5. Prediuresis

Mosquitoes previously fed on blood with trypanosomes (CUL98) were allowed to defecate and lay their eggs. Subsequently, they were allowed to feed on an anesthetized laboratory mouse. The feeding process was monitored, and fully fed mosquitoes were transferred into plastic tubes. Urine droplets were caught on the coverslip placed on the bottom of the plastic tube. Air-dried droplets were fixed by methanol, stained with Giemsa, and examined for the presence of trypanosomes. 

### 2.6. Wild Mosquito Collection and Identification of Parasites

Mosquitoes were trapped using CDC light traps (JW, Hock Company, Gainesville, FL, USA) without a bulb and baited with dry ice in 2018–2020 in Milovice forest, South Moravia, Czechia. Insects were collected in nylon nets connected to the traps, sorted according to species [25], and dissected. Dissected guts were examined under the light microscope for the presence of trypanosomes (infection status, intensity, and parasite localization). A part of the positive guts was used for the cultivation of kinetoplastids, and the rest was stored in ethanol for the barcoding of parasites (see below).

### 2.7. Wild Bird Studies

Bird sampling was done at three localities between May and July, as described in [14]. Adults and yearlings were mist-netted at watering places in a game reserve (Milovice forest, South Moravia, 48.821274, 16.693175) or in cooperation with registered ringers contributing to the program Constant Effort Site (CES), organized by the Prague Ringing Centre (Zeměchy, 50.231783, 14.272371 and Choteč, 49.999069, 14.280239 in Central Bohemia). Trapping and sampling were done by licensed workers according to national law and experimental guidelines.

### 2.8. Diagnostic PCR in Mosquitoes and Birds

DNA from mosquitoes’ guts and wild birds’ blood was extracted using a High Pure PCR Template Preparation Kit (Roche Diagnostic, Manheim, Germany) according to the manufacturer’s instructions and kept at −20 °C until further use. For parasite identification, trypanosome SSU rRNA was amplified using a specific nested PCR with S762 and S763 primers [26] for the first step and TR-F2 and TR-R2 [27] primers for the second step. For the identification of trypanosomes from cultures, single-step PCR with primers MedlinA and MedlinB [28] were used. All positive PCR products were purified with the enzymatic solution ExoSap (Thermo Scientific, Waltham, MA, USA), then sequenced at the core facility of the Faculty of Science. Sequences were examined in the program BioEdit and analyzed using the BLAST algorithm and nucleotide database NCBI.

### 2.9. Scanning Electron Microscopy

Guts positive for trypanosomes after experimental infection of mosquitoes were torn by insulin syringe then fixed in 2.5% glutaraldehyde in 5 mM HCl, 0.1 M cacodylate buffer for 24 h at 4 °C. Thereafter, they were processed at our core facility, the Laboratory of Electron Microscopy (https://www.natur.cuni.cz/biology/service/lem?set_language=en (accessed on 20 September 2021)), as follows: samples were post-fixed in 2% osmium tetroxide in the same buffer for 2 h at room temperature. After dehydration in a graded ethanol series, the guts were critical-point air-dried, sputter-coated with gold in a Polaron coater, and examined by the authors using a JEOL 6380LV scanning electron microscope. 

### 2.10. Light Microscopy and Measurement of Trypanosomes 

After the experimental infections, dissected mosquito guts were fixed with methanol on slides and stained with Giemsa stain, photographed at 1000x magnification with a CDC camera (DP70) using an Olympus BX51 microscope. Measurement of the cells was done using ImageJ software, and the data were processed using R software [29]

### 2.11. Animal Experimentation Guidelines

Animals were maintained and handled in the animal facility of Charles University in Prague following institutional guidelines and Czech legislation (Act No. 246/1992 and 359/2012 coll. on Protection of Animals against Cruelty in present statutes at large), which complies with all relevant European Union and international guidelines for experimental animals. All the experiments were approved by the Committee on the Ethics of Laboratory Experiments of the Charles University in Prague and were performed under permission of no. MSMT-31949/2019-5, MSMT-31949/2019-6, of the Ministry of Education, Youth and Sports and 50982/ENV/14-2961/630/14 of the Ministry of Environment. Investigators are certified for experimentation with animals by the Ministry of Agriculture of the Czech Republic.

## 3. Results

### 3.1. Experimental Infection of Mosquitoes

Two subspecies of *Cx. pipiens* were fed on blood with four strains of *T. thomasbancrofti* originating from three different hosts: CUL15, CUL98 (mosquito); OF19 (hippoboscid fly); and PAS343 (bird). Over 80% of *Cx. quinquefasciatus* and 70% of *Cx. molestus* were infected with isolate CUL98, with nearly 100% of them being heavy infections (Figure 1). Mosquitoes were also susceptible to isolate OF19 from the hippoboscid fly with more than 90% of *Cx. quinquefasciatus* and almost 70% of *Cx. molestus* being infected, and heavy infections were found in 98 and 89% of infected mosquito guts, respectively. In the case of isolate PAS343, more than 80% of *Cx. quinquefasciatus* and almost 50% of *Cx. molestus* were infected, with heavy infections prevailing as well. On the other hand, the susceptibility of mosquitoes to strain CUL15 was lower (28% of *Cx. molestus* and 5% of *Cx. quinquefasciatus* infected), and no heavy infections were detected except for one *Cx. quinquefasciatus*. With the exception of the CUL15 strain, infection rates were higher in *Cx. quinquefasciatus* (81–92%) compared to *Cx. molestus* (47–72%).

### 3.2. Localization of Trypanosomes in Mosquitoes

Trypanosomes in guts dissected 11- and 14-days post-infection (dpi) were localized in the hindgut and mainly formed rosettes (Figure 2 and Figure 3c). Starting from 20 dpi, changes in parasite localization and the appearance of unattached epimastigotes were observed (Figure 2 and Figure 3d). Trypanosomes were localized in the hindgut (Figure 3a–c); however, in some cases, they extended to the midgut. Only epimastigotes were found in the midgut. These free-swimming stages were also detected in the hindgut. Pure midgut infections or rosettes in midgut were not detected. For the dimensions of trypanosomes in mosquito guts, see Table 1. 

### 3.3. Experimental Transmission of Trypanosomes to Birds

Trypanosome strains PAS343, CUL98, and OF19—which were able to develop heavy infections in the mosquitoes’ guts—were used for the inoculation of laboratory birds. Birds were inoculated perorally or transconjunctivally with guts heavily infected by *T. thomasbancrofti*. All three strains of trypanosomes were infective for birds (Table 2). 

Canary 1 was positive between the 59th and 120th days after peroral inoculation (PAS343), then remained negative until a relapse of infection on days 580 and 665. Canary 2 was first positive on day 70 and last positive 141 days after peroral inoculation (OF19). Canary 3 was first positive two days after transconjunctival inoculation (CUL98) then remained negative until the last day of sampling. Zebra finch 5 was positive only on a single occasion on day 29 after inoculation. The identity of the parasites was confirmed by sequencing the SSU rRNA. All obtained sequences were identical with strains used for the experimental infection of mosquitoes. 

### 3.4. Transmission from Birds to Mosquitoes

Mosquitoes were allowed to feed on birds infected with *T. thomasbancrofti*. After dissection, 13% of the 23 *Cx. molestus* and 27% of the 47 *Cx. quinquefasciatus* fed on canary 1 were infected; similarly, 31% of the 29 *Cx. Molestus,* and 9% of the 11 *Cx. quinquefasciatus* fed on canary 2 were infected. Mosquitoes were allowed to feed on birds several times between the 60th and 120th days after inoculation. The first positive mosquitoes were detected on day 95 and the last positive on day 120 after inoculation. Trypanosomes were able to develop heavy infections localized in the hindgut, comparable to experimental infections as well as to infections from wild-caught mosquitoes (Figure 3g). The identity of the parasites was confirmed by sequencing SSU rRNA. Twenty-three mosquitoes fed on zebra finch 5 (35, 47, and 52 dpi) remained negative. 

### 3.5. Prediuresis

Stages of *T. thomasbancrofti* were observed on Giemsa-stained droplets of urine from prediuresis. Trypanosomes were observed as epimastigote forms in 5 out of 12 examined droplets (Figure 3e,f).

### 3.6. Prevalence of T. thomasbancrofti in Wild-Caught Mosquitoes

Between 2018 and 2020, 1367 wild-caught mosquitoes were dissected, of which 771 belonged to the genus *Culex*. The rest were mosquitoes of the genera *Aedes* (*n* = 592)*, Anopheles* (*n* = 15)*, Culiseta* (*n* = 28), and *Mansonia* (*n* = 24). Out of the 771 dissected *Culex* mosquitoes, 49 individuals (6.2%) were infected with kinetoplastids. *T. culicavium* had the highest prevalence, with 35 positive individuals (4.5%). Twelve mosquitoes were infected with monoxenous kinetoplastids, and a single individual with *T. thomasbancrofti* (0.13%). One *Cx. pipiens* had a mixed infection, and one harbored the mammalian species *T. theileri.* All species of mosquitoes infected by kinetoplastids were tested, but avian trypanosomes were identified exclusively in *Culex* mosquitoes. 

### 3.7. Prevalence of T. thomasbancrofti in Wild Passerines

In passerines screened between 2014 and 2019, *T. thomasbancrofti* was detected using blood cultivation in 13 (0.44%) of the 2943 sampled individuals. Infected species included Eurasian reed warbler (*Acrocephalus scirpaceus*), common chiffchaff (*Phylloscopus collybita*), wood warbler (*P. sibilatrix*), barn swallow (*Hirundo rustica*), sand martin (*Riparia riparia*), and Eurasian blackcap (*Sylvia atricapilla*). Prevalence in these species was 2.9% (*n* = 446), and generic prevalence was 1.7% (*n* = 762). The negative sampled bird species/genera included *Parus* spp., *Coccothraustes coccothraustes*, *Emberiza* spp., *Turdus* spp., *Carduelis* spp., *Ficedula albicollis*, *Fringilla coelebs*, *Sitta europaea*, *Passer montanus*, *Sturnus vulgaris*, *Erithacus rubecula*, and *Certhia* spp. (in descending order according to numbers; only those with at least 15 sampled individuals were included). Blood smears of birds positive for *T. thomasbancrofti* were inspected under the microscope at magnification x1000 for 10 min and the whole smear area at x200; a single trypomastigote was found (Figure 3h).

## 4. Discussion

Mosquitoes are notorious vectors of multiple pathogens (viruses, bacteria, protozoa, nematodes); however, their importance as trypanosome vectors is relatively unexplored, perhaps overshadowed by the transmission of *Plasmodium*. A trypanosome known to be transmitted by mosquitoes is *Trypanosoma rotatorium* from frogs [30]; mosquitoes were included with certainty among the vectors of avian trypanosomes much later [3,19]. Avian trypanosomes found in mosquitoes in our earlier studies were usually localized on the stomodeal valve, resembling the suprapylarian *Leishmania* in sandflies; their transmission, however, occurred by vector ingestion and not by bite as in sandflies. These trypanosomes belong to a species described as *T. culicavium* [19]. In this paper, we described the experimental life cycle of *T. thomasbancrofti* [24], which belongs to a different trypanosome group related to *T. avium* [3], and for which experimental evidence of transmission and a description of development in mosquitoes was lacking. Unlike *T. culicavium*, infections by *T. thomasbancrofti* are localized in the hindgut, similar to parasites found in *Aedes aegypti* [10,21].

For our experimental work, we used several isolates of *T. thomasbancrofti* to test their potential to develop infections in two *Cx. pipiens* subspecies, *Cx. quinquefasciatus* and *Cx. molestus*. These isolates originated from mosquitoes, a bird, and a hippoboscid fly. Surprisingly, the isolate from the hippoboscid fly was able to develop high infection rates, while one of the isolates from mosquitoes produced low infection rates and intensities. Strains differ in their performance; therefore, conclusions concerning vectorial competence should not be based on a single strain–vector combination. Moreover, strains with higher developmental plasticity (lower vector specificity) may have the potential to bridge different vertebrate hosts or even set up new vector–parasite combinations. Nevertheless, the role of hippoboscid flies as vectors of *T. avium* group in nature remains unclear. Due to the difficulty of the laboratory handling of hippoboscid flies, the development of infective stages in hippoboscids could not be tested. Hippoboscids spend most of their active lives on the host, and the blood is permanently present in their intestine, enabling nonspecific parasites to thrive and be isolated [3].

The localization of *T. thomasbancrofti* was in the hindgut, typical for trypanosomes belonging to *T. avium* s. l. (group C) [3,14,17]. Similar to other species from this group, after the rupture of the peritrophic matrix, trypanosomes migrate to the hindgut and rectum, where they stay attached by hemidesmosomes to the hindgut chitinous lining, as seen in *Simulium* spp. and *Ae. aegypti* [12,21]. However, in the course of infection, free and unattached epimastigotes appeared in the hindgut, and in some cases also in the midgut. We assume that these free stages are metacyclic forms infective for birds; guts with these forms were therefore used for the experimental inoculation of birds.

To test mosquitoes’ potential as vectors of *T. thomasbancrofti*, we experimentally inoculated laboratory birds. Despite different localization of *T. culicavium* and *T. thomasbancrofti* in mosquito guts, we showed that the route of transmission was identical (i.e., peroral). *T. thomasbancrofti* localization in hindgut however opens another potential way of transmission: transconjunctival. Bloodsucking Nematocera excrete excessive fluids during feeding on the host (prediuresis) [31], and metacyclic kinetoplastids can be present in these droplets [32]. Because mosquitoes feed readily around the eyes, prediuresis can play an important role in the life cycle of *T. thomasbancrofti* as well as in *T. avium* transmitted by black flies [12]. The number of trypanosomes in the droplets was significant (approximately 400 cells), and the morphology corresponded to putative metacyclic forms. 

Inoculated canaries repeatedly tested positive, either using blood cultures or xenodiagnosis. This natural way of mosquito infection further confirms the vectorial role of mosquitoes in nature. A zebra finch was positive only once, similar to the single canary from the study of Svobodová et al. [17] inoculated with *T. bennetti*. The sudden appearance of parasites in the peripheral blood can be associated with prolonged photoperiod, stress caused by breeding, or experimental manipulation, as demonstrated for trypanosomes [33]. A relapse after two years from inoculation confirms that infection by avian trypanosomes is chronic with an intermittent appearance in peripheral blood [15,21,34]. 

The prevalence of *T. thomasbancrofti* in wild-caught mosquitoes in Central Europe is low [23,35]. Only 0.9% of *Culex pipiens* trapped at raptor nests in Czechia harbored infections in the hindgut compared to 8% localized on the stomodeal valve; out of the 28 trypanosomatid strains established, only one belonged to *T. thomasbancrofti* [23]. Mosquitoes caught using CO_2_ in our study showed a low prevalence of *T. thomasbancrofti* as well. The prevalence of *T. culicavium* was 3.6%, while that of *T. thomasbancrofti* was 0.1%. In avian hosts, the prevalence of *T. thomasbancrofti* was low as well: only 0.4% of screened passerines were positive. Since the trypanosome is transmitted by ingestion, it is appropriate to focus on insectivorous avian species; then, the prevalence increases to nearly 3%. Therefore, it can be seen that parasites circulate in specific hosts, and the prevalence obtained by screening for unspecific hosts may be misleading. 

## 5. Conclusions

Our study confirms that (1) *Culex* mosquitoes are highly susceptible to *T. thomasbancrofti* infections; (2) trypanosomes can develop metacyclic stages in mosquito guts that are transmissible to birds perorally or transconjunctivally; (3) the prevalence of *T. thomasbancrofti* among wild mosquitoes and birds was lower than the prevalence of *T. culicavium*; and (4) the complete life cycle of the *T. thomasbancrofti* was achieved, and mosquitoes can thus be considered as confirmed vectors.

## Figures and Tables

**Figure 1 microorganisms-09-02101-f001:**
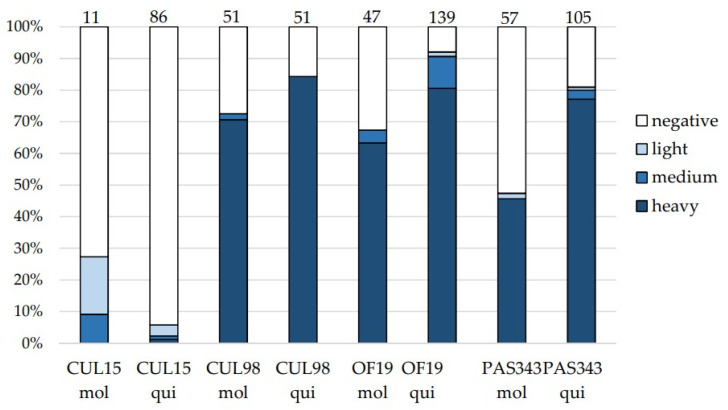
Infection rates and intensities of infection in mosquitoes *Culex molestus* (mol) and *Culex quinquefasciatus* (qui) experimentally infected with *Trypanosoma thomasbancrofti* strains CUL15, CUL98, OF19, and PAS343. Infection intensities: light, 1–100 parasites; medium, 100–1000 parasites; heavy, >1000 parasites per gut. Numbers of dissected females are presented above the columns.

**Figure 2 microorganisms-09-02101-f002:**
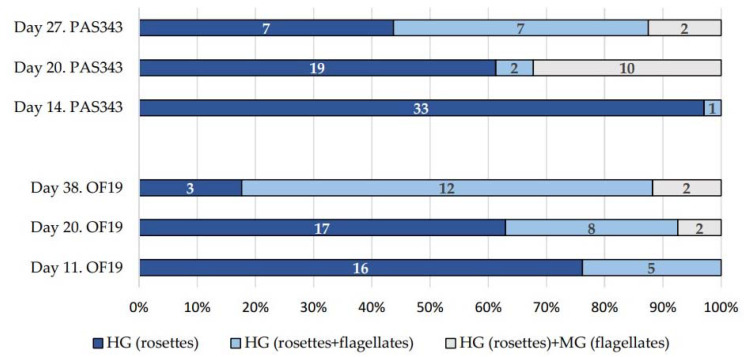
Changes of trypanosome localization in *Cx. quinquefasciatus* guts experimentally infected with isolate OF19 and PAS343. The numbers of dissected females are shown in the columns. HG, hindgut; MG, midgut.

**Figure 3 microorganisms-09-02101-f003:**
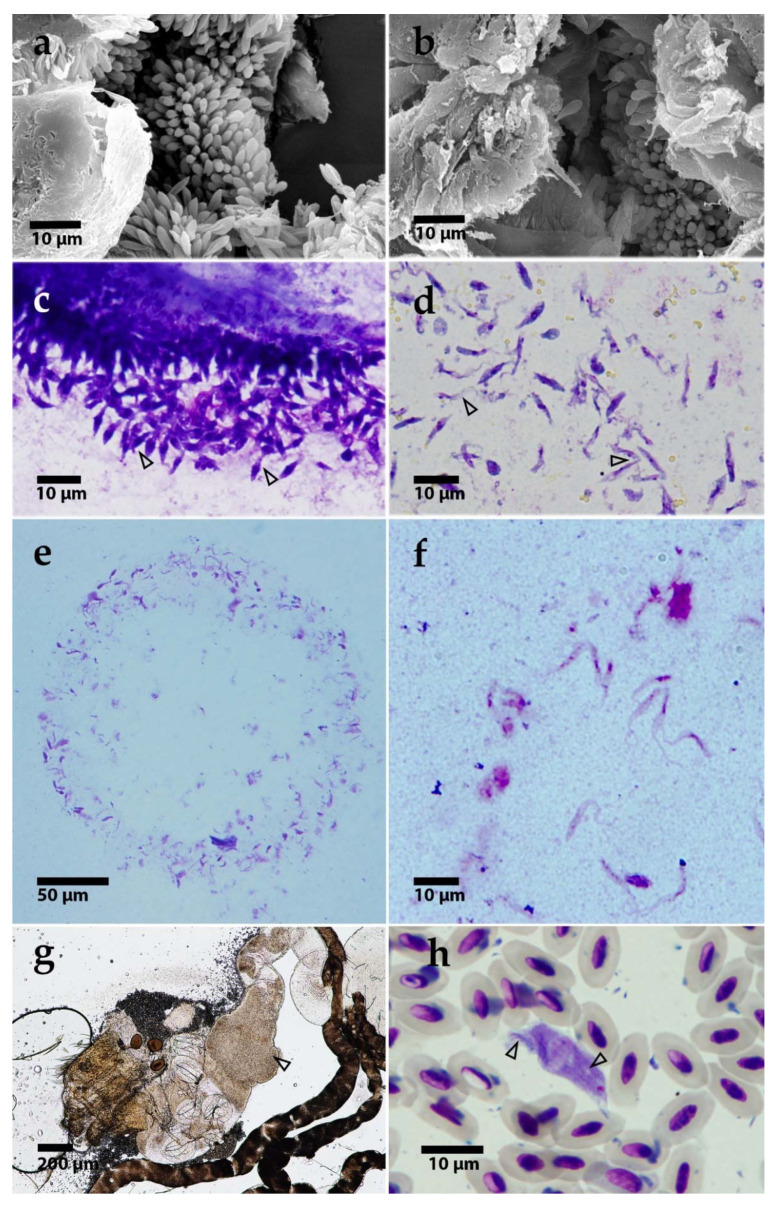
(**a**,**b**) Scanning electron microscopy of *T. thomasbancrofti* OF19 (**a**) and PAS343 (**b**) in the gut of *Culex quinquefasciatus* after experimental infection (**c**,**d**). Light microscopy of trypanosome morphotypes in *Culex quinquefasciatus* gut: rosettes (**c**) and epimastigotes (**d**). Prediuresis droplet with numerous trypanosomes (CUL98) (**e**). A detail of epimastigotes from prediuresis droplet (**f**). Dissected *Culex* mosquito gut with trypanosomes after xenodiagnosis; arrowhead pointing to the mass of parasites (**g**). *T. thomasbancrofti* trypomastigote from barn swallow (*Hirundo rustica*) blood with visible striation (see arrows for striation) (**h**).

**Table 1 microorganisms-09-02101-t001:** Morphometry of trypanosomes in mosquito gut. Values in micrometers. SE, standard error. At least 30 cells were measured for each strain/host combination.

Cell Type	Body Length Mean ± SE (Range)	Body Width Mean ± SE (Range)	Flagellum Length Mean ± SE (Range)
Epimastigote	8.7 ± 0.2 (4.5–14.6)	1.2 ± 0.0 (0.6–2.0)	7.8 ± 0.2 (3.0–13.2)
Rosette	7.2 ± 0.1 (3.5–11.3)	2.2 ± 0.0 (1.1–3.6)	1.6 ± 0.1 (0–8.8)

**Table 2 microorganisms-09-02101-t002:** Results of bird inoculations. Birds were inoculated perorally (po) or transconjunctivally (tc) with *T. thomasbancrofti* from the laboratory-reared mosquitoes *Cx. molestus* (mol) and *Cx. quinquefasciatus* (qui).

Bird	Strain	Dose	Mosquito Strain	Infection Route	Result	Day First Positive	Day Last Positive	Day Last Checked
Canary 1	PAS343	8 guts	qui	po	positive	59	665	790
Canary 2	OF19	7 guts	qui	po	positive	70	141	720
Canary 3	CUL98	7 guts	qui	tc	positive	2	2	160
Canary 4	CUL98	7 guts	qui	tc	negative			160
Zebra finch 5	PAS343	8 guts	mol	po	positive	29	29	620
Zebra finch 6	PAS343	8 guts	qui	po	negative			520

## Data Availability

Not applicable.

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
