# Peer review of "Complete Life Cycle of Trypanosoma thomasbancrofti, an Avian Trypanosome Transmitted by Culicine Mosquitoes"

_microorganisms, 2021, doi:10.3390/microorganisms9102101_

Round 1

Reviewer 1 Report

My compliments.  This is a clear and well-presented piece of work.  The English usage is generally so good that where it has slight shortcomings I have annotated the MS (as attached).  I hope you will accept this without offence!

I have only very small scientific queries, which again you will find as comments in the annotated MS, and only one possible error.  This involves a scale bar/magnification query in the micrographs.

Author Response

Dear reviewer,

we appreciate precious time in reviewing our paper and providing valuable comments. We have been able to incorporate changes to reflect suggestions provided by you.

For membrane feeding, we really use all the blood for inactivation and not just serum, information has been added to the methodology.
Furthermore, the correct scale bars have been added to the images, thank your reminder.

We hope you find the revised manuscript acceptable for publication. Thank you once again for your consideration.

Yours Sincerely, 
Magdaléna Fialová 

Reviewer 2 Report

Comments to Authors

The aim of this manuscript is to confirm the experimental transmission of T. thomasbancrofti to birds, as well as its localization in vectors. Authors performed complex experimental infections with parasite-canaries-mosquitoes. Also, the natural prevalence in mosquitoes and birds were determined.

The manuscript is original, well planned and written. The authors investigated T. thomasbancrofti and determined its life cycle.

I have only a few questions below

The authors conducted experimental infections with mosquitoes with 4 strains of T. thomasbancrofti. In figure 2 and paragraph 3.2. (Localization of trypanosomes in mosquitoes) authors gave results only from one strain (PAS343). Why authors didn’t investigate other T. thomasbancrofti strains?

Did the authors check the infection intensities in guts before inoculation to birds?

Line 39 – parasitelife should be parasite life

Author Response

Dear reviewer,

we appreciate precious time in reviewing our paper and providing valuable comments. We have been able to incorporate changes to reflect suggestions provided by you.

The authors conducted experimental infections with mosquitoes with 4 strains of T. thomasbancrofti. In figure 2 and paragraph 3.2. (Localization of trypanosomes in mosquitoes) authors gave results only from one strain (PAS343). Why authors didn’t investigate other T. thomasbancrofti strains?

  • We have added a graph with isolate OF19. Other strains were not tested (CUL 15- not enough infected mosquitoes, CUL 98 – all positive mosquitoes were used for inoculation, prediuresis experiments and measurement of trypanosomes.

Did the authors check the infection intensities in guts before inoculation to birds?

  • in the methodology we mentioned that guts used for inoculation of birds were infected with different strains of T. thomasbancrofti. To determine the infection status, were guts dissected and examined under a microscope (see  2.2. and 2.3.)  

Line 39 – parasitelife should be parasite life

  • Revised

We hope you find the revised manuscript acceptable for publication. Thank you once again for your consideration.

Yours Sincerely, 
Magdaléna Fialová 

Reviewer 3 Report

Dear authors,

The manuscript Complete life cycle of Trypanosoma thomasbancrofti, an avian trypanosome transmitted by culicine mosquitoes presents an interesting study that was done investigating vector capacity of different culicine species for the transmission of the avian parasite T. thomasbancrofti. The authors also managed to investigate the prevalence of this parasite in wild birds and wild mosquitoes from different genera.

My major comment is related to the presentation of the methodology used in the study. Since several experiments were conducted and some of them depended on the result of others, it would be better that the authors present all the criteria used in the methods sections and not in the results sections. For example, they only mentioned that mosquitoes with high infections were used in the bird experiment in the results, and not on methods.

Following, I present my minor comments point-by-point:

Line 13: When the scientific name starts a new sentence, it should be spelled, please, check along with the text.

Lines 15-16: please, include the rates of infection of isolate OF19.

Line 23: nature should not be in capital letter

Line 25: should be ‘prediuresis’ and not ‘presiuresis’

Line 39: should be ‘parasite life’

Line 42: I suggest including the citation of Bernotiene et al. (2020) - https://doi.org/10.1016/j.actatropica.2020.105555. Since there are not many groups working with avian trypanosomes, and even less conducting experimental research, I believe this would contribute to the knowledge in this field.

Line 79: what was the humidity that mosquitoes were kept? Please, include. Did the authors use 50% sucrose solution for any specific reason? It is more common to use a 10% solution.

Line 89-96: how many birds were used in this experiment? Which methods were used to screen the birds before the experiment? Why the authors used only blood cultivation for the diagnosis of infection? Which mosquitoes species were used to infect birds? Were the birds infected in pairs, as for the mosquito experiment? Which lineages were used to infect each bird species? How the authors selected the way of infection (perorally or transconjunctival)? Please, provide more detailed information.

Line 96: gene names should be written in italic. Please, check along with the text

Line 98: how many mosquitoes were in the cages during experimental infections? All positive birds were used in this experiment? How these mosquitoes were kept during the experiment? Please, include information.

Line 104: Why only isolate CUL98 was used in the prediuresis experiment?

Line 110: I suggest saying ‘wild mosquito collection’

Line 117: The barcoding methods were for mosquitoes or for parasites?

Line 118: I suggest saying ‘wild bird studies’

Line 119: suggest saying ‘wild bird’. Please, include more information on where these birds were sampled, maybe GPS coordinates.

Line 136: should be in italic

Line 137: This protocol was used for laboratory-reared or wild mosquitoes? Please, specify. Is this a new protocol? If not, include the reference.

Line 143: Were these mosquito guts intact or were they smashed in the slide before? Please, clarify.

Line 166: please spell ‘resp.’

Line 167: authors are mentioning that ‘more than 50% of Cx. molestus were infected with heavy infection’, however, figure 1, it is showing that it was less than 50%. Please, review this information.

Line 180: Figure 3 is mentioned before figure 2, please check. Say ’20 dpi’ not ‘dpi 20’.

Figure 2: replace day, by dpi. What do HG and MG mean? Please, include it in the legend. Scientific names should be written in italic, please, check along with the text, figure legends, and tables. Why are the authors showing only the results of isolate 343, what about the other isolates? Did they have similar results?

Line 205: Transconjunctivally or trans-conjunctivally (line 195)?

Table 2: I believe that ‘pos’ means positive and ‘neg’, negative. Please, include the explanation.

Line 207: should be in italic

Line 208: say ‘laboratory-reared mosquitoes’

Line 211: all mosquitoes in all birds were fed during this period? Did the authors compare both ways of transmission, if one of them was more effective than the other?

Lines 222: Were all the mosquitoes genera tested for Trypanosoma? Please, clarify.

Line 230: Which other bird species were tested? Please include this information, maybe in a supplementary file. Which time of the year these birds were tested?

Lines 236-238: It could be moved to the methods section.

Line 246: Please, include the indication of the striation in the figure.

Line 291: What does ‘cca’ means?

Lines 313-318: I suggest presenting the numbers between brackets and use comma and semicolons along with the sentence.

Round 2

Reviewer 3 Report

Dear authors,

Thank you for addressing all my comments and suggestions. 

Congratulations on the work done.